# Identification of Phytoplankton-Based Production of the Clam *Corbicula japonica* in a Low-Turbidity Temperate Estuary Using Fatty Acid and Stable Isotope Analyses

**Dongkyu Seo, Changseong Kim, Jaebin Jang, Dongyoung Kim and Chang-Keun Kang ***

School of Earth Sciences and Environmental Engineering, Gwangju Institute of Science and Technology, Gwangju 61005, Republic of Korea; dkseo9832@gm.gist.ac.kr (D.S.); changseong@gist.ac.kr (C.K.); wkdwoqls7@gist.ac.kr (J.J.); dongyoung@gwnu.ac.kr (D.K.)
* Correspondence: ckkang@gist.ac.kr

**Abstract:** The brackish water clam, *Corbicula japonica*, acts as an ecosystem engineer in estuaries. To identify its resource-use patterns in the low-turbidity temperate Seomjin River estuary of Korea, we analyzed stable isotope and fatty acid (FA) biomarkers to differentiate allochthonous and autochthonous dietary sources, and examined the effects of clam size, salinity gradient, and season. The $\delta^{13}C$ and $\delta^{15}N$ values were consistent across the three factors. The $\delta^{13}C$ values of the clams were similar to those of both riverine- and estuarine-suspended particulate organic matter (R- and E-SPOM), while their $\delta^{15}N$ values were 2–4‰ higher, indicating an equal contribution of both sources to the clam diet. Biogeochemical proxies and FA compositions of SPOM indicate that estuarine phytoplankton significantly contribute to the E-SPOM pool. Moreover, the similarity in FA profiles between *Corbicula* and E-SPOM indicates that phytoplankton-derived organic matter is the primary source of nutrition for the clam, with minimal impact from growth, salinity gradient, or seasonal changes. Our study suggests that in low-turbidity estuaries with high phytoplankton production, allochthonous organic matter makes a negligible contribution to clam nutrition, compared to high-turbidity estuaries. This finding could provide insights into the variations in the trophic structure of estuarine food webs across diverse regions.

**Keywords:** *Corbicula japonica*; dietary source; stable isotopes; fatty acids; low turbidity; Seomjin River estuary

## 1. Introduction

*Corbicula* is an infaunal bivalve clam genus that plays an important role as an ecosystem engineer in the functioning of brackish and/or freshwater ecosystems [1,2]. The Asian brackish water clam (*Corbicula japonica*, *Prime*, 1864) lives in the soft bottom sediment of estuaries where it feeds on suspended particles. The genus acts as a biological filter through removing considerable amounts of suspended material, which increases estuaries' water clarity and nutrient mineralization rates [3,4] Its ingestion of sedimentary particles and production of pseudofeces may also affect nutrient dynamics [5]. The genus can also be a valuable biological monitor of ecosystem change or pollution [6]. *Corbicula* species have high secondary production [7,8]; they are used as fishery resources for human consumption [9]. Much attention has been paid to the species' trophic ecology to elucidate ecological processes in *Corbicula*-dominated brackish water ecosystems, such as material cycling.

*C. japonica* is a nonselective, suspension-feeding bivalve that filters particles—including phytoplankton, bacteria, and nonliving detritus—using an inhalant siphon above the sediment surface [9]. *C. japonica* also uses its foot for deposit-feeding on the sediment surface, collecting bacteria in substrate sediment and deposited organic matter such as debris from higher plants [10]. Some small bivalves (e.g., *C. fluminea* and *Mysella bidentata*) employ the foot to collect sedimentary particles as they become adults, whereas some large bivalves

confine the use of the foot to assist suspension-feeding during their larval and late juvenile stages [11]. These feeding characteristics may result in ontogenetic diet changes between juvenile and adult phases (e.g., see *Cerastoderma edule* and *Macoma balthica*; [12,13]). Thus, it may be expected that the *C. japonica* would change the feeding zone from water column (suspension-feeding) to sediment surface (deposit-feeding) with size, due to age.

The food resources assimilated by the *Corbicula* species may vary, depending on environmental conditions. Kasai et al. [9] and Yamanaka et al. [10] found the utilization of benthic microalgae and bacteria by deposit-feeding varied in response to spatial variation of different food resources. Way et al. [5] also reported that *C. fluminea* can deposit-feed, especially when the concentrations of suspended particles are insufficient and organic substrates are abundant. High seasonal precipitation can affect particle concentration and composition in stream water [14]. The resulting shorter retention time of water in fast-flowing streams induces a larger inflow of terrestrial sources of organic matter to the brackish water zone, compared to a prevalence of autochthonous sources during low discharge periods [15]. Along the northeast Asian coast, about two-thirds of annual precipitation is concentrated during the summer monsoon period, affecting carbon discharge through streams [16]. Based on the likely seasonality in the availability of organic matter sources, it might be expected that food sources for *C. japonica* would vary from pre- to post-monsoon seasons. As a result, the documented sources of variation in the resource utilization of *C. japonica* may generate a lack of generality in its resource utilization.

Direct observation of food consumption in the field is impractical over long periods. Gut-content estimations of diet composition or fecal pellet analyses are likely to be valid only for recent feeding activities, over seconds to hours. Different digestibility of different foods makes identification of what is ingested and also assimilated, difficult. Furthermore, the traditional analysis methods are labor- and time-demanding steps. As alternatives to the traditional methods, the use of biochemical tracers, such as stable isotopes (SIs) and FAs, have been more effective in determining the food sources of feeding organisms [17]. These chemical tracers reflect the assimilation of putative food sources into body tissues, providing dietary information over longer periods (weeks to months) [18–20]. Distinct isotope signatures of organic matter sources are conserved through the trophic steps [17]. Carbon isotope ratio ($\delta^{13}$C) values of animal tissues reflect those of their diets, with little trophic-step fractionation (<1.0‰), thus identifying animals' carbon sources [21,22]. In contrast, nitrogen isotope ratio ($\delta^{15}$N) values of animal tissues tend to be on average 3.4‰ (2–4‰) higher than those of the source diets, but are still able to act as proxies of animals' trophic positions [23,24]. FAs also have been used as dietary biomarkers to trace trophic transfer from producers to consumers. Different primary producers (e.g., marsh phanerogams, macroalgae, riverine and marine microalgae) have different FA profiles (especially those of essential FAs, [25]), which can be assimilated in a conservative manner [20,26]. Because de novo FA synthesis in feeding organisms is limited, dietary fats can be distinguishable from nondietary fats, using FA analysis [27,28].

Our goal was to identify the relative importance of allochthonous (terrestrial) versus autochthonous (the estuary itself) sources of organic matter to the major nutritional sources of *C. japonica* in the Seomjin River estuary, Korea (Figure 1). The availability of the river-borne organic matter may depend on the seasonal discharge of river water. The water column of this estuarine habitat is characterized by exceptionally low concentrations of suspended particulate matter (SPM) and low turbidity [29]. Given the ontogenetic development of the siphon in *C. japonica* and the hydrological conditions in the Seomjin estuary, we anticipated that the nutritional sources of *C. japonica* would vary between allochthonous (terrestrial) and autochthonous (the estuary itself) sources, with clam size due to age and season. We measured seasonal SI and FA compositions in *C. japonica* and in four putative sources of organic matter. These were R-SPOM, coastal and estuarine phytoplankton, benthic microalgae, and the cordgrass *Phragmites australis*, which is the dominant macrophyte of the Seomjin tidal flat. We hypothesized that if *C. japonica* uses different dietary sources with ontogeny and season, its SI and FA compositions would make

consistent changes with those of putative sources of organic matter. To test this hypothesis, we examined whether the SI and FA compositions of *C. japonica* vary with (1) ontogeny (growth effect), (2) space (site effect by salinity gradient), and (3) season (pre-, post-, and non-monsoon). Our final goal was to unravel how the unique hydrological features of this estuary determine the trophism of *C. japonica* and the further trophic base of the entire food web in the estuarine system.

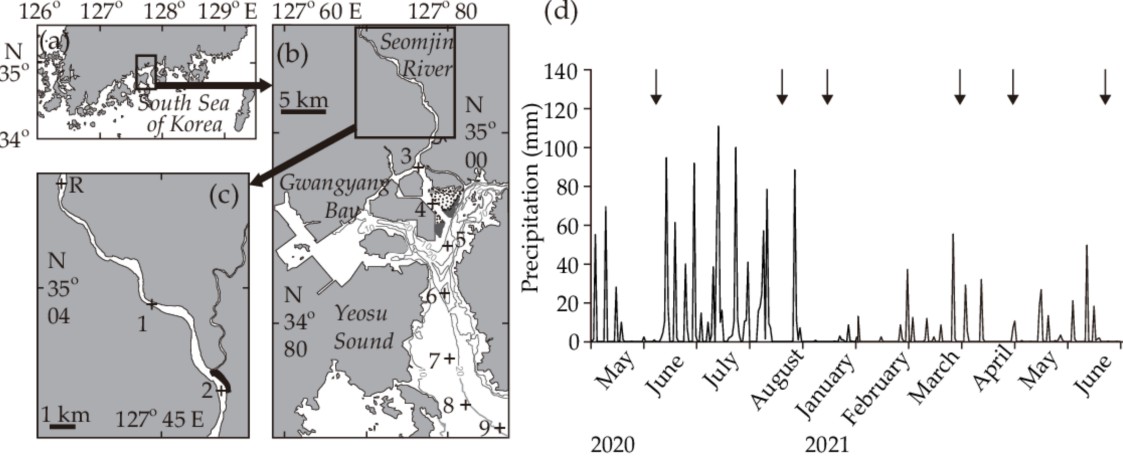

**Figure 1.** (**a**) Map showing the location of Gwangyang Bay; (**b**) map showing the location of the sampling sites in the bay; (**c**) map showing the sampling sites (1, 2, and River) in the estuarine channel. The dark area represents the *P. australis* bed. Nine stations for the monitoring of environmental conditions and biological communities by Korean Long-term Marine Ecosystem Research are indicated by the crosses and numerals; (**d**) sampling dates during the pre- and post-monsoon periods (2020) and sampling dates in three non-monsoon seasons (2021). Arrows indicate the sampling dates.

## 2. Materials and Methods

### 2.1. Study Area

This study was conducted in the Seomjin River estuary on the southern coast of Korea (Figure 1a), which constitutes the main habitat of *C. japonica* and serves as an important fishing ground (over 90% of domestic *Corbicula* catch). The Seomjin River is the only large river that flows directly to the seacoast with no interruption due to estuarine dams, in South Korea. The river empties a large volume of freshwater into Gwangyang Bay and is connected to the sea through the bay channel to the south. The river discharge exhibits considerable seasonal variability with an annual average of 120–160 m$^3$ s$^{-1}$ (30–95 m$^3$ s$^{-1}$ in the winter dry season and 300–400 m$^3$ s$^{-1}$ in the summer monsoon; [30]). Therefore, the quantity and composition of dissolved inorganic nutrients, phytoplankton, and SPM show clear seasonal variations [29], affecting spatial and seasonal availability of different organic matter sources that form the trophic base of the food web [31]. Broad plains of the common reed *P. australis* occupy the upper littoral zone from the lower reach of the river to the estuarine channel [32]. Green/brown carpets of microphytobenthos (mainly benthic diatoms) cover the sandy mudflat of the littoral zone, which is free of macrophytes [32].

Considering the seasonal recruitment of *Corbicula* spats in June and the monsoon climate of this region (Figure 1b), samplings were conducted in June–August 2020 to measure SIs and FAs of *Corbicula* and its putative food sources in the lower reach of the Seomjin River and the narrow estuarine channel. Because of overlapping values in SIs between sources (see Results), additional samplings were conducted in January, March–April, and June 2021 to test seasonal variation in FA compositions of *Corbicula* and its putative food sources, and differences between sources at the same sites. Specimens of *C. japonica* were sampled at two sites along the estuarine channel. Putative food sources of *Corbicula* on each sampling occasion, riverine SPM, estuarine SPM, phytoplankton, and *Phragmites* were sampled at respective sites (a riverine site and two estuarine sites, respectively).

### 2.2. Sample Collection and Laboratory Processing

On each sampling occasion, water temperature and salinity were recorded using a conductivity-temperature-depth meter (CTD; Sea-Bird Electronics, Bellevue, WA, USA). For clam collection, sediments were taken using a 0.12-m$^2$ van Veen grab, and sieved using a 0.5 mm mesh net. All sizes of *C. japonica* were collected and stored in sterilized 1 L plastic bottles. For isotope determination of SOM, sediment was collected by scraping the sediment surface into 50 mL conical tubes. For determinations of SIs and FAs of SPM, approximately 60 L of water was collected using a van Dorn water sampler at the two clam collection sites and a river site at the lower reach. Water was filtered using a 200 μm mesh screen to remove large particles and zooplankton. *P. australis* leaves were collected by hand in the adjacent littoral zone. Phytoplankton for isotope determination were collected using a 20 μm mesh net and filtered through a 200 μm mesh screen to remove large particles and zooplankton. All samples were preserved with dry ice and transported to the laboratory immediately.

For determinations of total SPM and chlorophyll *a* (Chl *a*), 1 L of seawater for each measurement was filtered through previously weighed Whatman GF/F filters (47 mm, pore size = 0.7 μm), assisted by a gentle vacuum (150–200 mm Hg). For measurements of suspended particulate organic carbon (POC) and particulate nitrogen (PN), 500 mL of water was filtered using previously combusted Whatman GF/F filters. Filters containing SPM were dried at 60 °C for 72 h and the SPM mass was computed from the weight difference before and after filtration. Chl *a* was extracted with 90% acetone for 24 h in the dark at −20 °C and quantified using a fluorometer (Turner Designs, Sunnyvale, CA, USA) following the method of Holm-Hassen et al. [33]. The C and N elemental composition was analyzed using an elemental analyzer (Eurovector 3000 Series; Eurovector, Milan, Italy).

After freeze drying, small quantities of SPM and sediment samples destined for δ$^{13}$C analysis were subsampled, and acidified to remove inorganic carbonates. This was carried out by adding 1–2 drops of 1 M HCl for the SPOM samples and enough HCl until bubbling stopped for the sedimentary organic matter (SOM) samples.

Clams were sorted by size and cleaned with Milli-Q water. Biometric measurement for their shell lengths were made using a Vernier caliper. Flesh tissue was dissected using a surgical knife and was stored at −70 °C.

Finally, all flora, fauna, and SPOM samples were lyophilized and pulverized to a fine powder using a ball mill (Retsch MM200 mixer mill; Hyland Scientific, Stanwood, WA, USA).

### 2.3. Stable Isotope Analysis

Filters containing SPOM were wrapped in tinfoil disks, and powdered samples were packed in tinfoil capsules. The samples were oxidized in the Eurovector elemental analyzer at 1030 °C, then a stream of helium gas was used to convey the resultant $CO_2$ and $N_2$ to a continuous-flow isotope ratio mass spectrometer (Isoprime; GV Instruments, Manchester, UK). SI ratios were determined relative to the conventional standards of Vienna PeeDee Belemnite for carbon and atmospheric $N_2$ for nitrogen, and expressed in delta (δ) notation, according to the following equation:

$$\delta X(‰) = \left[ \left( R_{\text{sample}} / R_{\text{standard}} \right) - 1 \right] \times 10^3,$$

where X is $^{13}$C or $^{15}$N, and *R* is the $^{13}$C/$^{12}$C and $^{15}$N/$^{14}$N ratio for carbon and nitrogen. International standard reference materials (USGS-24 for carbon and IAEA-N1 for nitrogen) were used to calibrate δ$^{13}$C and δ$^{15}$N values. Analytical precision of the overall analysis and preparation was approximately 0.1‰ for δ$^{13}$C and 0.2‰ for δ$^{15}$N.

### 2.4. Fatty Acid Analysis

Lipid extractions of *C. japonica*, *P. australis*, and SPOM followed the Bligh and Dyer method [34]. Subsequently the FA compositions of samples were analyzed as FA methyl esters (FAMEs) prepared using the American Oil Chemists' Society method. After evapo-

rating chloroform with nitrogen gas, lipids were saponified with 1.5 mL of 1.5 N NaOH (in MeOH) at 100 °C. Saponified lipids were transferred into FAMEs by adding 2 mL of 14% $BF_3$/MeOH and gently shaken at 100 °C in sealed vials under nitrogen. After cooling, 2 mL of isooctane was added and nitrogen gas was charged. After adding 5 mL of saturated NaCl solution, nitrogen gas was charged again. When the layers were separated, the upper isooctane layer was transferred by pipette to a vial. A gas chromatograph (GC; Agilent Technologies, Santa Clara, CA, USA) connected to a flame ionization detector was used to analyze FAMEs. FAME classes were separated using a flexible, fused silica capillary column (bonded carbowax, 0.25 μm film thickness and 30 m × 0.25 mm internal diameter). The carrier gas was nitrogen. The temperature of the GC was programmed to rise from 50 °C to 250 °C, at 30 °C min$^{-1}$ to 150 °C, then at 2 °C min$^{-1}$ to 250 °C, followed by 10 min at 250 °C. For FAME identification, peak retention times were compared with FAME standards mixtures (Supelco 37 Component FAME Mix and 18919-1AMP; Sigma-Aldrich, Bellefonte, PA, USA).

*2.5. Statistics*

Standard statistical analyses were conducted with a SPSS Statistics 21.0a package (IBM Corp., Armonk, NY, USA), while for more sophisticated analyses a permutational multivariate analysis of variance (PERMANOVA), nonmetric multidimensional scaling (nMDS), and linkage tree analysis (LINKTREE, as described later) were performed, using PRIMER 6 and PERMANOVA+ (PRIMER-E, Plymouth, UK). Before the formal analyses, data sets were tested for normality and homogeneity of variances using the Shapiro–Wilk and Levene's tests, respectively. Multivariate analysis of variance (MANOVA) was performed to test the significance of the isotopic difference between sources of organic matter. When significance was detected, a two-way analysis of variance (ANOVA) followed by a Tukey post hoc test was conducted to test whether $\delta^{13}C$ and $\delta^{15}N$ values differed significantly between sites and seasons for E-SPOM, SOM, phytoplankton, and *C. japonica*. Student's *t*-test was used to identify significant differences in $\delta^{13}C$ and $\delta^{15}N$ values between seasons for *P. australis* and R-SPOM. The threshold for statistical significance was set at $p < 0.05$.

We selected 12 FA biomarkers of C16:0, C18:0, C22:0, C13:0 + C15:0 + C17:0, C16:1ω7, C18:2ω6, C18:3ω3, C18:3ω6, C20:4ω6, C20:5ω3, C22:6ω3, and the ω3:ω6 ratio. Individual FA classes in each sample were expressed by percentage proportion of total FAs. This choice was based on the literature data of major FA biomarkers of phytoplankton, vascular plants, detritus, riverine (terrestrial) matter, and bacteria (Table S1). In most bivalves, de novo synthesis of polyunsaturated FAs (PUFAs) is essential for their biological activities [35], but is strictly limited [36]. Therefore, PUFAs can be used to identify the contributions of putative sources of organic matter to their diets (Table S2). Although bivalves synthesize saturated FAs (SFAs) and monounsaturated FAs (MUFAs) in their flesh tissues [37], unique classes of SFAs and MUFAs in each source may also be used as biomarkers to assess the source contributions to the bivalve nutrition.

Before statistical analysis, FA data were converted using an arcsine square root formation [26]. We employed PERMANOVA to test which factors made significant differences between putative food sources (E-SPOM, R-SPOM, and *P. australis*), based on the main factors, that is, source type and season, and for *C. japonica*, based on the three factors of clam size, site, and season. Because there was an interaction between source type and season among putative food sources (see the Results), pairwise PERMANOVA within the season was also conducted. Two-way ANOVA with Tukey's honestly significant difference test identified the significance of each biomarker. If there were a significance among groups in terms of a characteristic of *C. japonica*, the Kruskal–Wallis test and the following pairwise comparisons using the Mann–Whitney *U* test identified the significance of each biomarker according to its non normality and homogeneity of variances (Shapiro–Wilk and Levene's tests).

We used nMDS to visualize the FA patterns among putative food sources and *C. japonica*, based on Euclidean distance matrixes [38]. A multivariate regression tree developed from Classification and Regression Trees [39], LINKTREE, was also applied to yield a hierarchical dendrogram discriminating different assemblages. We set the minimum group size to one and the minimum split size to three. FA data for LINKTREE were not transformed before analysis because homogeneous covariance matrixes and normality are not required [40].

## 3. Results

### 3.1. Environmental Conditions

Water temperature showed a broad annual range from a winter minimum (6.9 °C at estuarine site 1 in January) to a summer maximum (25.5 °C at the riverine site in August), typical of temperate zone ranges (Figure 2). Salinity was 0 at the riverine site, remained relatively low, at 0.1, at site 1, and fell within a narrow range between 1.9 (March 2021) and 8.3 (April 2021) at site 2, during the study period. Chloroph1ll (Chl) *a* concentrations were relatively low and constant, with a range of 0.32–1.0 μg L$^{-1}$ at the riverine site compared to 0.33–5.03 μg L$^{-1}$ at estuarine sites 1 and 2, where concentrations were highly variable but with no marked seasonal or spatial patterns. SPM concentrations ranged from 2.4 mg L$^{-1}$ to 9.4 L$^{-1}$ at the riverine site and 4.0 mg L$^{-1}$ to 9.7 mg L$^{-1}$ at sites 1 and 2, with no obvious seasonal or spatial trends. The percentage of SPOM/SPM was low at the riverine site (22.3–67.1) compared to the estuarine sites 1 and 2 (51.5–90.5). POC and PN concentrations showed similar ranges between the riverine site (156–303 μg L$^{-1}$ and 23–45 μg L$^{-1}$, respectively) and the estuarine sites 1 and 2 (96–272 μg L$^{-1}$ and 17–64 μg L$^{-1}$, respectively). Molar C:N and POC:Chl *a* ratios were relatively high (8.1–9.6 and 514–845, respectively), at the riverine site compared to those at estuarine sites 1 and 2 (2.3–6.6 and 70–392, respectively), except for values in March and April 2021.

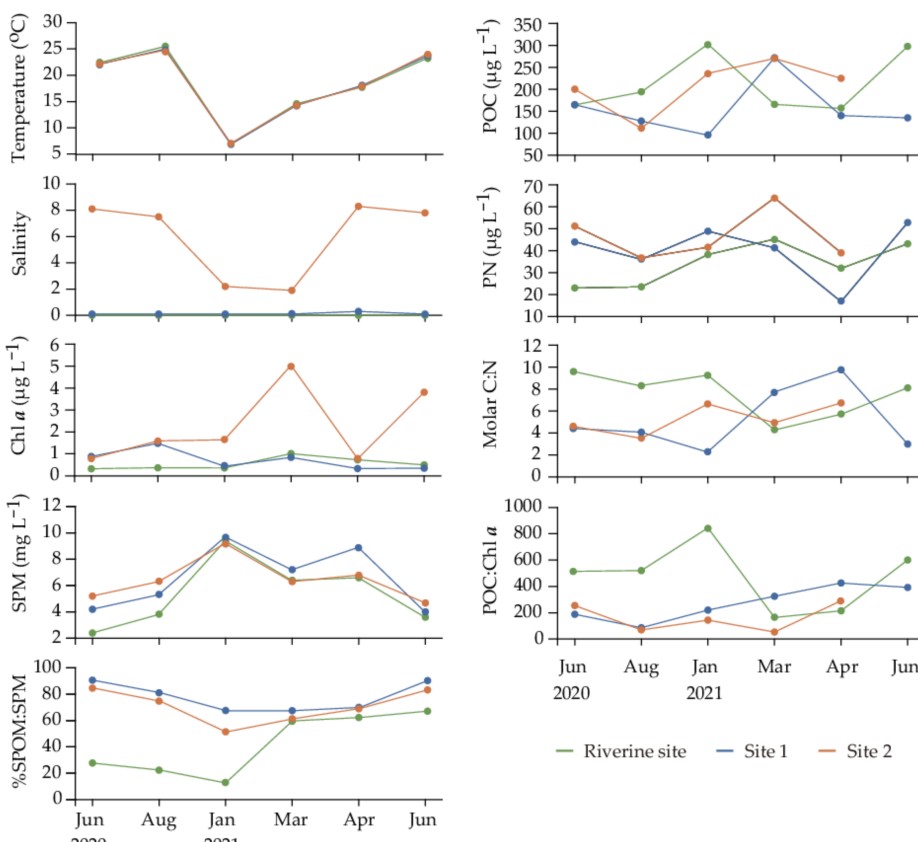

**Figure 2.** Environmental conditions at three sites on each sampling occasion. C:N, POC to PN in the suspended matter; %SPOM:SPM, the fraction (%) of SPOM of SPM; POC:Chl *a*, POC to Chl *a* in the suspended matter.

### 3.2. δ13C and δ15N Values of Putative Food Sources

No seasonal effects on $\delta^{13}$C values of food sources were detected (analyses of variance [ANOVA], $p > 0.05$ for E-SPOM, SOM, and phytoplankton; Student's *t*-test, $p > 0.14$ for *p. australis* and R-SPOM), except for phytoplankton (ANOVA, $F_{1,4} = 25.0$, $p = 0.007$) (Table 1). There was a significant site effect in $\delta^{13}$C values for E-SPOM and phytoplankton ($F_{1,4} = 10.14$ and $56.93$, $p = 0.034$ and $0.002$, respectively), but the effect was minor, at <1.5‰. Regarding source $\delta^{15}$N values, a significant but small difference (<1‰) in E-SPOM was exhibited between sites, but no significant differences between sites or seasons were observed for the other sources.

**Table 1.** Two-way ANOVA (E-SPOM, SOM, and phytoplankton), student *t*-test (*P. australis* and R-SPOM), and means for $\delta^{13}$C and $\delta^{15}$N values of putative food sources.

| Food Sources | | Site | | *P* (within Source) | | |
|---|---|---|---|---|---|---|
| | | 1 | 2 | Site | Season | Interaction |
| | | $\delta^{13}$C (‰) | | | | |
| E-SPOM | | $-25.3 \pm 0.2$ [b,c] (4) | $-24.7 \pm 0.3$ [c,d] (4) | 0.034 * | 0.136 | 0.999 |
| SOM | | $-24.2 \pm 0.2$ [d] (8) | | 0.885 | 0.058 | 0.769 |
| Phytoplankton | June | $-26.0 \pm 0.3$ [b] (2) | $-24.7 \pm 0.1$ [c,d] (2) | 0.002 * | 0.007 * | 0.999 |
| | August | $-25.1 \pm 0.2$ [b,c] (2) | $-23.9 \pm 0.3$ [d] (2) | | | |
| *P. australis* | | $-27.7 \pm 0.2$ [a] (4) | | | 0.327 | |
| R-SPOM | | $-25.3 \pm 0.4$ [b,c] (4) | | | 0.141 | |
| One-way ANOVA (among sources) | | $F_{8,23} = 42.00$; $p < 0.001$ * | | | | |
| | | $\delta^{15}$N (‰) | | | | |
| E-SPOM | | $4.8 \pm 0.2$ [i] (4) | $5.7 \pm 0.6$ [j,k] (4) | 0.022 * | 0.368 | 0.411 |
| SOM | | $3.5 \pm 0.2$ [h] (8) | | 0.452 | 0.219 | 0.654 |
| Phytoplankton | | $5.2 \pm 0.3$ [i,j] (8) | | 0.999 | 0.064 | 0.763 |
| *P. australis* | | $6.0 \pm 0.2$ [k] (4) | | | 0.149 | |
| R-SPOM | | $5.0 \pm 0.3$ [i,j] (4) | | | 0.076 | |
| One-way ANOVA (among sources) | | $F_{5,26} = 25.00$; $p < 0.001$ * | | | | |

Notes: Means followed by the different superscript letters are significantly different ($p < 0.05$, Tukey post hoc test). Asterisks indicate significance at $p < 0.05$.

The MANOVA test denoted significant differences in the isotopic signatures in the five putative sources of organic matter (i.e., R-SPOM, E-SPOM, SOM, phytoplankton, *P. australis*; Pillai's trace: $F_{4,27} = 16.70$, $p < 0.001$). A subsequent ANOVA test revealed significant differences in both $\delta^{13}$C and $\delta^{15}$N values between sources ($F_{8,23} = 42.0$ and $F_{5,26} = 25.0$, $p < 0.001$, Table 1). The mean $\delta^{13}$C values of food sources ranged from $-27.7 \pm 0.2$‰ (*P. australis*) to $-23.9 \pm 0.3$‰ (phytoplankton at site 2 in August). E-SPOM had similar $\delta^{13}$C values ($-25.3 \pm 0.2$‰ and $-24.7 \pm 0.3$‰) to R-SPOM ($-25.3 \pm 0.4$‰) (Tukey post hoc test, $p > 0.05$), overlapping with the phytoplankton range: $-26.0 \pm 0.3$ ‰ to $-23.9 \pm 0.3$‰, depending on seasonal and spatial variations. The mean $\delta^{13}$C value of SOM ($-24.2 \pm 0.2$‰) was close to those of SPOM and phytoplankton. The mean $\delta^{15}$N values of food sources fell within a range from $3.5 \pm 0.2$‰ (SOM) to $6.0 \pm 0.2$‰ (*P. australis*). The remaining sources had intermediate $\delta^{15}$N values between the two extremes (SOM and *P. australis*; Tukey post hoc test, $p > 0.05$).

### 3.3. δ13C and δ15N Values of C. japonica

Regarding isotopic variations in the soft tissues of clams, the determination was first made among size groups (range: 2–15 mm and 4–25 mm in shell length at sites 1 and 2) collected in June 2020 (Figure 3a,b; Table S3). Younger individuals (spats) settled in late spring would be expected to grow rapidly during the summer; therefore, we failed to collect young individuals in August. No clear trends in the $\delta^{13}$C and $\delta^{15}$N values relating

to clam size were apparent (coefficients of variation of 1.2% and 1.1% for $\delta^{13}$C values, and 7.1% and 3.9% for $\delta^{15}$N values at sites 1 and 2, respectively). The mean $\delta^{13}$C values of *C. japonica* at the two estuarine sites were similar, falling within narrow ranges from $-25.7 \pm 0.01$‰ to $-24.7 \pm 0.2$‰ (overall mean, $-25.3 \pm 0.3$‰) at site 1, and $-25.0 \pm 0.2$‰ to $-24.3 \pm 0.1$‰ ($-24.7 \pm 0.3$‰) at site 2. Mean $\delta^{15}$N values ranged from $7.2 \pm 0.7$‰ to $9.2 \pm 0.02$‰ (overall mean, $8.4 \pm 0.6$‰) and $7.2 \pm 0.1$‰ to $8.0 \pm 0.5$‰ ($7.7 \pm 0.3$‰) at sites 1 and 2, respectively.

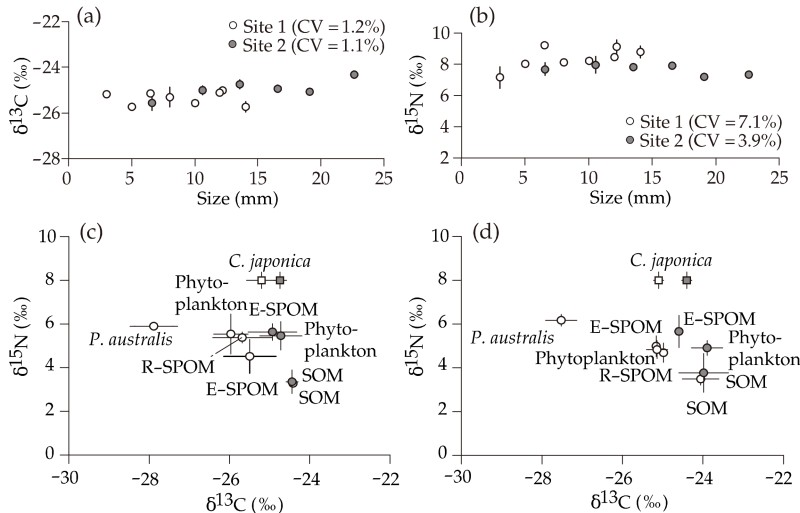

**Figure 3.** (**a**) $\delta^{13}$C values according to size groups of *C. japonica* collected in June 2020; (**b**) $\delta^{15}$N values according to size groups of *C. japonica* collected in June 2020. White and gray symbols represent site 1 and site 2, respectively. *N* = 4 for each size group; (**c**) biplots for $\delta^{13}$C and $\delta^{15}$N values of *C. japonica* (square) and putative food sources (circle) in June; (**d**) biplots for $\delta^{13}$C and $\delta^{15}$N values of *C. japonica* (square) and putative food sources (circle) in August. White symbols represent values at site 1 and gray symbols represent values at site 2.

A two-way ANOVA revealed significant differences in the mean $\delta^{13}$C values of *C. japonica* (12–15 mm individuals) between sites ($F_{1,20} = 47.14$, $p < 0.001$) and seasons ($F_{1,20} = 5.215$, $p = 0.033$), with no interaction ($F_{1,20} = 2.675$, $p = 0.118$) (Table 2). The mean $\delta^{13}$C values were consistently higher at site 2 than at site 1 (Tukey post hoc test, $p < 0.05$) and in August than in June ($p < 0.05$). However, the differences were no more than 0.7‰ between sites and 0.2‰ between seasons. In addition, no significant spatial and seasonal differences were observed in their $\delta^{15}$N values (overall mean, $8.0 \pm 0.3$‰; $F_{1,20} = 2.054$ and 3.297, $p = 0.167$ and 0.084, respectively).

**Table 2.** Two-way ANOVA and means for $\delta^{13}$C and $\delta^{15}$N values of *C. japonica*.

| Factor | $\delta^{13}$C | | | | $\delta^{15}$N | | | |
|---|---|---|---|---|---|---|---|---|
| | df | MS | F | p | df | MS | F | p |
| Site | 1 | 1.729 | 47.14 | <0.001 * | 1 | 0.160 | 2.054 | 0.167 |
| Season | 1 | 0.191 | 5.215 | 0.033 * | 1 | 0.257 | 3.297 | 0.084 |
| Interaction | 1 | 0.098 | 2.675 | 0.118 | 1 | 0.030 | 0.379 | 0.545 |
| Residual | 20 | 0.037 | | | 20 | 0.071 | | |
| | | Site 1 | | Site 2 | | | Overall mean | |
| June | | $-25.2 \pm 0.3$ [a] (8) | | $-24.7 \pm 0.1$ [b] (4) | | | $8.0 \pm 0.3$ (24) | |
| August | | $-25.1 \pm 0.1$ [a] (8) | | $-24.4 \pm 0.1$ [b] (4) | | | | |

Notes: Means followed by the different superscript letters are significantly different ($p < 0.05$, one-way ANOVA with Tukey post hoc test). Asterisks indicate significance at $p < 0.05$.

While the $\delta^{13}$C values of *C. japonica* were roughly proportional with those of R- and E-SPOM, and of phytoplankton on each sampling occasion, their $\delta^{15}$N values were 2–4‰

higher than those of the three sources of organic matter (Figure 3c,d). Considering the common fractionation factors (+2 to +4‰) in $\delta^{15}N$ values per trophic level and *C. japonica* $\delta^{13}C$ values distinct from *P. australis* (this study), the combination of $\delta^{13}C$ and $\delta^{15}N$ values in the clams and their putative food sources indicated the importance of R- and E-SPOM (including phytoplankton) for the diets of *C. japonica*. However, close proximity between R- and E-SPOM on the $\delta^{13}C$–$\delta^{15}N$ biplots impeded the identification of their relative importance for *C. japonica* nutrition. For this reason, we further analyzed the FA compositions of *C. japonica* and its putative food sources.

### 3.4. Fatty Acid Compositions of Putative Food Sources

Individual FA-class profiles for the three putative food sources we analyzed showed significant variations between source types and seasons (Table S4). Despite significant interactions between both factors, FA profiles for putative food sources were consistently dominated by SFA compounds with higher proportions in R-SPOM (monthly means, 46.6–90.5%) than those in E-SPOM (38.6–49.6%) and *P. australis* (37.0–60.6%). Proportions of MUFA compounds were comparable, and relatively low in all food sources (16.8–31.9% for E-SPOM, 6.5–19.1% for R-SPOM, and 7.5–28.3% for *P. australis*). In contrast, the importance of PUFAs was much greater in E-SPOM (25.4–44.5%) and *P. australis* (31.9–39.2%) than in R-SPOM (3.0–36.7%).

Although interaction effects between both factors were also observed in most of the FA classes (11 of 12), their importance varied among source types (Table S4; see also Table S1). The most common FAs were C16:0, C18:0, C20:5ω3, and C22:6ω3, C18:3ω3, and C18:2ω6 in E-SPOM. In contrast, R-SPOM was characterized by high proportions of C16:0, C18:0, and C18:2ω6. The most common FAs in *P. australis* were C16:0, C18:2ω6, C18:0, and C18:3ω6. The ω3:ω6 ratio was much higher in E-SPOM (0.9–2.8) than those in R-SPOM (0–0.8) and *P. australis* (0.0–0.5) ($p < 0.05$). Based on 11 FA biomarkers and the ω3:ω6 ratio, two-way PERMANOVA revealed significant differences in FA profiles between seasons (Pseudo-$F_{4,20}$ = 17.748, $p < 0.001$) and source types (Pseudo-$F_{2,20}$ = 40.812, $p < 0.001$) (Table 3). Given the strong interaction (Pseudo-$F_{8,20}$ = 10.911, $p < 0.001$) between both factors, further pairwise comparisons within seasons showed significant differences between source types, except for similar FA profiles of R-SPOM and *P. australis* in January and June 2021 ($p = 0.339$ and 0.328). The nMDS ordination indicated that the FA profiles of E-SPOM were separated from those of R-SPOM and *P. australis* (Figure 4). While E-SPOM was placed on the lower and left side on the plane, R-SPOM and *P. australis* were scattered on the right side. This separation was attributed to close correlations of E-SPOM with ω3:ω6 ratio, C22:6ω3, C20:4ω6, and C20:5ω3, and R-SPOM and *P. australis* with C22:0, C18:2ω6, and C18:0.

**Table 3.** Two-way PERMANOVA results of differences in FA biomarker compositions (food sources). Tested factors are season (August 2020, January 2021, April 2021, and June 2021) and source type (E-SPOM, R-SPOM, and *P. australis*), and their interaction for food sources.

| Group | Factors | df | Sums of sqs | Mean sqs | *Pseudo-F* | *p* (perm) |
|---|---|---|---|---|---|---|
| | Season | 4 | 3462 | 1154 | 17.75 | <0.001 * |
| | Source type | 2 | 5307 | 2653 | 40.81 | <0.001 * |
| Food sources | Interaction | 8 | 4256 | 709.3 | 10.91 | <0.001 * |
| | Residual | 20 | 780.2 | 65.01 | | |
| | Residual | 1 | 7.114 | 7.951 | | |
| Pairwise "Season × Source type" for pairs of levels of factor "Source type" | | | | | | |
| Within "Season" | | df | Sums of sqs | Mean sqs | *Pseudo-F* | *p* (perm) |
| August 2020 | E-SPOM vs. R-SPOM | 1 | 848 | 848 | 770.8 | <0.001 * |
| | | 2 | 2.200 | 1.100 | | |
| | E-SPOM vs. *P. australis* | 1 | 1414 | 1413 | 15.03 | <0.001 * |
| | | 2 | 0.181 | 0.094 | | |
| | R-SPOM vs. *P. australis* | 1 | 1011 | 1010 | 883.7 | <0.001 * |
| | | 2 | 2.289 | 1.144 | | |

**Table 3.** *Cont.*

| Group | Factors | df | Sums of sqs | Mean sqs | *Pseudo-F* | *p* (perm) |
|---|---|---|---|---|---|---|
| January 2021 | E- SPOM vs. R-SPOM | 1 | 863.4 | 863.4 | 4.772 | <0.001 * |
| | | 2 | 361.5 | 180.9 | | |
| | E-SPOM vs. *P. australis* | 1 | 813.8 | 813.8 | 5.591 | <0.001 * |
| | | 2 | 291.1 | 145.57 | | |
| | R-SPOM vs. *P. australis* | 1 | 898.7 | 898.7 | 15.86 | 0.339 |
| | | 2 | 113.4 | 56.68 | | |
| Mar–April 2021 | E- SPOM vs. R-SPOM | 1 | 364.4 | 364.4 | 14.71 | <0.001 * |
| | | 2 | 49.55 | 24.78 | | |
| | E-SPOM vs. *P. australis* | 1 | 1471 | 1470 | 86.70 | <0.001 * |
| | | 2 | 33.93 | 16.96 | | |
| | R-SPOM vs. *P. australis* | 1 | 533.2 | 533.2 | 28.92 | <0.001 * |
| | | 2 | 36.88 | 18.44 | | |
| June 2021 | E- SPOM vs. R-SPOM | 1 | 2175 | 2175 | 19.99 | <0.001 * |
| | | 2 | 217.6 | 108.8 | | |
| | E-SPOM vs. *P. australis* | 1 | 2394 | 2394 | 31.82 | <0.001 * |
| | | 2 | 150.5 | 75.25 | | |
| | R-SPOM vs. *P. australis* | 1 | 1558 | 1558 | 10.36 | 0.328 |
| | | 2 | 300.8 | 150.4 | | |

Notes: Asterisks indicate significance at *p* < 0.05.

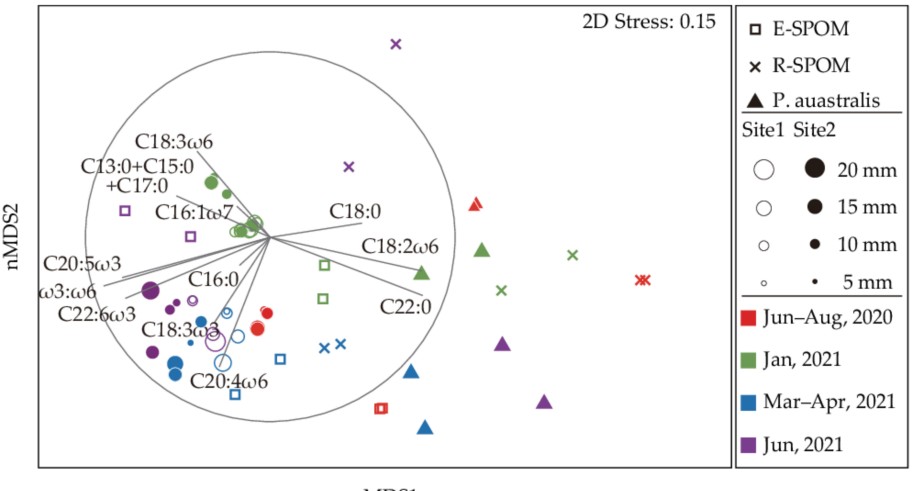

**Figure 4.** nMDS plot for 11 FA biomarkers and the ω3:ω6 ratio of *C. japonica* and its putative food sources (E-SPOM, R-SPOM, and *P. australis*) from June 2020 to June 2021.

*3.5. Fatty Acid Compositions of C. japonica*

FA profiles for *C. japonica* were consistently dominated by SFA, which varied monthly from 41.3% (April 2021) to 67.0% (June 2020) (ANOVA, *p* = 0.010, Table 4). The monthly varying importance of PUFA (15.8–40.8%) and MUFA (14.3–27.2%) was also observed (ANOVA, *p* = 0.004 and < 0.001, respectively). Despite considerable seasonal variations, the most common FAs were typically C16:0, C18:0, C20:5ω3, C22:6ω3, and C16:1ω7. Based on 11 FA biomarkers and the ω3:ω6 ratio, a three-way PERMANOVA revealed a seasonal effect (Pseudo-$F_{4,21}$ = 20.2, *p* < 0.001), but no effects of site (*p* = 0.354) or clam size (*p* = 0.128) on FA profiles of *C. japonica*. In addition, there were no interactions (Table 5).

The nMDS plot displayed the association in the FA profiles between food sources (i.e., E-SPOM, R-SPOM, and *P. australis*) and *C. japonica* (Figure 4). Despite seasonal variation in the FA profiles, *C. japonica* individuals of different sizes collected at both sites were closely associated with E-SPOM on the left side of the plane. In contrast, R-SPOM and *P. australis* were on the right side. The FAs with the greatest effect on sample ordination were the ω3:ω6 ratio, C22:0, C18:2ω6, C20:5ω3, C22:6ω3, and C20:4ω6. The nMDS1 was strongly

correlated with the ω3:ω6 ratio (correlation coefficients, $r = -0.901$), C22:0 ($r = 0.826$), C18:2ω6 ($r = 0.822$), C20:5ω3 ($r = -0.803$), and C22:6ω3 ($r = -0.787$), which separated *C. japonica* and E-SPOM from R-SPOM and *P. australis*. The nMDS2 presented monthly differences in *C. japonica* and was closely associated with C20:4ω6 ($r = -0.706$), C18:3ω3 ($r = -0.469$), and C18:3ω6 ($r = 0.459$).

**Table 4.** FA composition (% of total FAs) of *Corbicula japonica* from June 2020 to June 2021 and results of analysis of variance.

| FA | June 2020 (2) | August (2) | January 2021 (12) | April 2021 (8) | June 2021 (8) | *p* |
|---|---|---|---|---|---|---|
| C13:0 + C15:0 + C17:0 | 5.4 ± 0.7 [b] | 2.9 ± 0.0 [b] | 2.0 ± 0.9 [a] | 1.7 ± 0.5 [a] | 3.7 ± 0.8 [b] | 0.001 * |
| C16:0 | 40.0 ± 0.4 [c] | 29.4 ± 0.1 [bc] | 18.8 ± 2.7 [a] | 21.3 ± 3.5 [a] | 24.2 ± 6.9 [ab] | 0.019 * |
| C18:0 | 13.2 ± 0.1 [c] | 5.8 ± 0.1 [ac] | 8.0 ± 1.1 [a] | 6.5 ± 0.4 [b] | 7.6 ± 2.6 [ab] | 0.005 * |
| C22:0 | — | — | — | 0.2 ± 0.3 | 0.1 ± 0.2 | 0.228 |
| ∑ SFA | 67.0 ± 0.1 [c] | 47.1 ± 0.9 [bc] | 44.6 ± 7.7 [ab] | 41.3 ± 3.2 [a] | 54.8 ± 3.9[c] | 0.010 * |
| C16:1ω7 | 7.4 ± 0.0 [c] | 9.2 ± 0.2 [c] | 4.3 ± 0.7 [b] | 3.5 ± 3.3 [abc] | 1.8 ± 1.4 [a] | 0.008 * |
| ∑ MUFA | 16.1 ± 0.2 [ab] | 27.2 ± 0.7 [bc] | 25.4 ± 3.6 [b] | 17.9 ± 3.3 [a] | 14.3 ± 1.6 [ac] | 0.004 * |
| C18:2ω6 | 1.7 ± 0.0 [bc] | 3.7 ± 0.1 [c] | 1.5 ± 0.7 [b] | 1.7 ± 1.1 [b] | 0.4 ± 0.5 [a] | 0.004 * |
| C18:3ω3 | 2.4 ± 0.0 [ab] | 1.7 ± 0.0 [a] | — | 2.7 ± 0.5 [b] | 1.6 ± 0.9 [a] | 0.021 * |
| C18:3ω6 | 0.7 ± 0.1 [a] | 0.3 ± 0.0 [a] | 8.9 ± 1.6 [c] | 1.6 ± 0.1 [b] | 1.8 ± 0.8 [ab] | <0.001 * |
| C20:4ω6 | 2.3 ± 0.1 | 3.8 ± 0.1 | — | 3.3 ± 0.4 | 3.4 ± 0.9 | 0.143 |
| C20:5ω3 | 3.7 ± 0.1 [a] | 8.8 ± 0.0 [ab] | 6.9 ± 1.2 [b] | 12.6 ± 2.3 [c] | 7.1 ± 3.3 [b] | 0.001 * |
| C22:6ω3 | 3.4 ± 0.1 [a] | 4.6 ± 0.1 [a] | 7.9 ± 1.2 [b] | 16.4 ± 4.9 [c] | 13.9 ± 2.6 [c] | <0.001 * |
| ∑ PUFA | 15.8 ± 0.1 [a] | 25.6 ± 0.2 [ab] | 29.9 ± 4.2 [b] | 40.8 ± 5.1 [c] | 30.9 ± 4.2 [b] | <0.001 * |
| ω3:ω6 ratio | 2.0 ± 0.1 [a] | 1.9 ± 0.0 [a] | 1.9 ± 0.3 [a] | 4.8 ± 1.2 [b] | 3.9 ± 1.2 [b] | <0.001 * |

Notes: Means followed by the different superscript letters are significantly different ($p < 0.05$, Mann–Whitney U test) on each row. —, not detected. Asterisks indicate significance at $p < 0.05$. The number ($n$) of analytical samples is in parenthesis. ∑ SFA: a sum of C4:0, C6:0, C8:0. C10:0, C11:0, C12:0, C13:0, C14:0, C15:0, C16:0, C17:0, C18:0, C22:0, C21:0, C22:0, C23:0, and C24:0. ∑ MUFA: a sum of C15:1, C16:1ω7, C17:1, C18:1ω9t, C18:1ω9c, C20:1, C22:1n9, and C24:1. ∑ PUFA: a sum of C18:2ω6, C18:3ω3, C18:3ω6, C20:2, C20:3ω3, C20:3ω6, C20:4ω6, C20:5ω3, C22:2 and C22:6ω3. ω3: a sum of C18:3ω3, C20:3ω3, C20:5ω3, and C22:6ω3. ω6: a sum of C18:2ω6, C18:3ω6, C20:3ω6, and C20:4ω6.

**Table 5.** Three-way PERMANOVA results of differences in FA biomarker compositions (*C. japonica*). Tested factors are season (June 2020, August 2020, January 2021, April 2021, and June 2021), site (1 and 2), size (5–19 mm; varies from season and site), and their interaction. The size factor is nested in season and site factors, since size groups are not all the same in different seasons and sites.

| Group | Factors | df | Sums of sqs | Mean sqs | *Pseudo-F* | *p* (perm) |
|---|---|---|---|---|---|---|
| | Month | 4 | 4487.1 | 1121.8 | 20.22 | <0.001 * |
| | Site | 1 | 58.6 | 58.6 | 1.0612 | 0.354 |
| *C. japonica* | Month × Site | 4 | 210.41 | 52.603 | 0.94819 | 0.500 |
| | Size (Month × Site) | 21 | 1187.8 | 56.561 | 7.9506 | 0.128 |
| | Residual | 1 | 7.114 | 7.114 | | |

Notes: Asterisks indicate significance at $p < 0.05$.

LINKTREE classified patterns of the FA compositions of *C. japonica* and their putative food resources (Figure 5). The ω3:ω6 ratio clearly separated E-SPOM and *C. japonica* from *P. australis* and R-SPOM, except for *P. australis* in April 2021 and R-SPOM in June 2021. These exceptional cases of *P. australis* and R-SPOM had low proportions of C20:5ω3, showing a further separation from E-SPOM and *C. japonica*. Seasonal variations in the FA compositions of E-SPOM and *C. japonica* were characterized by proportions of essential FAs (C20:4ω6, C22:6ω3, C18:2ω6, and the ω3:ω6 ratio). In contrast, *P. australis* and R-SPOM tended to be separated by proportions of C22:0.

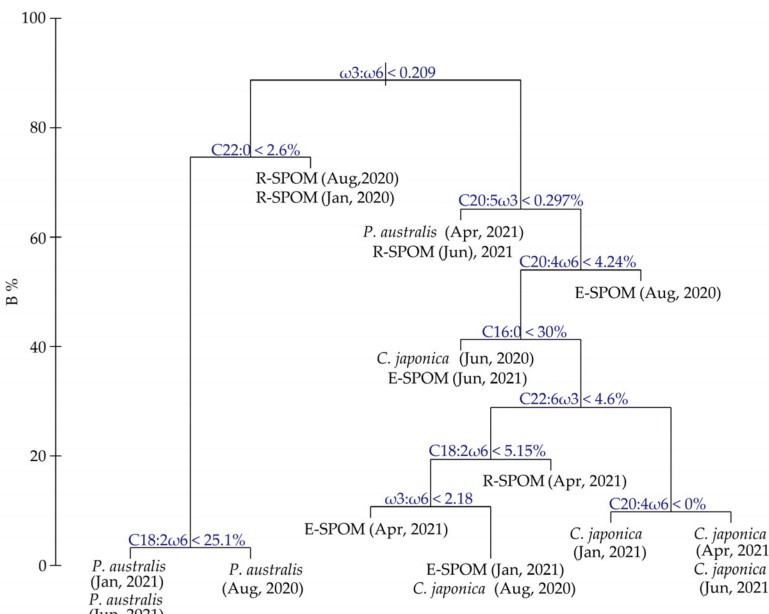

**Figure 5.** LINKTREE plot for putative food sources and *C. japonica*. B% is a separation scale between groups ranging from 0 to 100, which gives an absolute quantification of differences across separation. Criteria of FA compositions mean the threshold levels responsible for splits leading to left division.

## 4. Discussion

Previous isotope measurements revealed a change from R-SPOM to marine POM in the *C. japonica* diets along an estuarine gradient [41,42]. An isotope study in the river mouth of the Hyeongsan River estuary of Korea also concluded the trophic importance of R-SPOM for *C. japonica* diets [43]. In the present study, we found consistent $\delta^{13}$C and $\delta^{15}$N values in the clam tissues according to clam size, site, and season, which indicate a dominant contribution of a particular source to the clam diet. Both $\delta^{13}$C and $\delta^{15}$N values of R- and E-SPOM were close to each other in this river–estuary continuum [29,44]. *C. japonica* had very close $\delta^{13}$C values to both SPOM sources [31,44]. A lack of discrimination of isotopic values between the two sources led to an ambiguity in identifying the nutritional source of *C. japonica* in our river system. To overcome this ambiguity in SI values and to test our hypothesis, we applied FA composition to the identification of nutritional sources of *C. japonica*. Our results, based on FA profiles, further reject our initial hypothesis, instead suggesting that autochthonous phytoplankton production in the estuarine water column contributes more to the *C. japonica* nutrition, with no clear effects of clam ontogeny, location within estuary, or seasonal food availability. Our finding supports the fact that the trophic role of R-SPOM in higher-trophic-level organisms is limited to the riverine area, and is of little importance even to the upper-estuarine consumers.

### 4.1. Origin of Suspended Particulate Organic Matter

The similarity in $\delta^{13}$C values between R-SPOM and *C. japonica* allowed us to previously emphasize the significant importance of riverine (terrestrial) particulate matter for the nutrition of *C. japonica* [42,43]. A recent study has also reported the trophic importance of R-SPOM for zooplankton production in the Seomjin River estuary [45]. In the present study, R- and E-SPOM and *C. japonica* exhibited close homogeneity in their $\delta^{13}$C values, while *C. japonica* had 2.3–3.2‰ higher $\delta^{15}$N values than those of R- and E-SPOM, thus suggesting an important contribution of R-SPOM to the sources of E-SPOM and the *C. japonica* diet. However, the high disparity in quantity and biogeochemical proxies (i.e., molar C:N, POC:Chl *a*, %SPOM:SPM, and FA composition) of SPM between the riverine and estuarine waters suggests a low contribution of riverine POM to the estuarine POM pool. Further, FA biomarkers displayed a high dissimilarity between R- and E-SPOM but a close similarity

between E-SPOM and *C. japonica*. Our results from the biogeochemical proxies of SPM and FA biomarkers may allow us to reject the previous SI-based interpretation that the riverine POM serves an important role as a dietary resource of *C. japonica* in this low-turbidity estuarine system [43].

The proximity of isotope values between R- and E-SPOM may indicate that large-scale transport of riverine POM represents an important part of the E-SPOM pool. In turbid estuarine systems where riverine terrestrial particulate matter is dominant, the isotopic similarity between R-SPOM and consumers points to its nutritional importance for the upper estuarine food web [46–48]. However, the availability and nutritional role of R-SPOM may decrease in estuaries exhibiting minor riverine discharge [49]. SI values of phytoplankton can vary greatly, because of the isotopic composition of the dissolved inorganic carbon (DIC) pool, species-specific isotopic discrimination, and their growth rates in estuaries experiencing highly variable environmental conditions [50]. Indeed, in the low-turbidity estuarine condition of the Seomjin River estuary, the community composition of phytoplankton differs from that of the highly saline offshore zone [31], and their $\delta^{13}C$ values are very close to that of R-SPOM, due to conservative carbon mixing between the low concentration of DIC in freshwater and its high concentration in seawater [44]. Considering the high primary productivity of phytoplankton in the upper-estuarine zone [29], the contribution of R-SPOM to the estuarine POM pool—which determines its availability and role in the diets of *C. japonica*—should be reevaluated, as discussed below.

The SPM concentrations were very low (2.4–9.7 mg $L^{-1}$ in the present study); hence, the POC flux through the Seomjin River was estimated to be low (11–139 g C $s^{-1}$) compared to 772–1810 g C $s^{-1}$ ω in the highly turbid neighboring Geum River [51]. The short residence time of estuarine water (2.5 d, [52]) restricts the retention of riverine SPM in the estuary. A combined effect of low SPM quantity and rapid water exchange allows the estuarine water to sustain low turbidity, which promotes high in situ production of phytoplankton in a eutrophic condition, compared with highly turbid estuaries [29]. The high %SPOM:SPM ratio (51.5–90.5 compared to 12.6–67.1 of riverine SPM) suggests an increased contribution of autochthonous phytoplankton production to the SPOM pool of the studied estuarine system, which is supported by a low C:N ratio (2.3–6.6 compared to 8.1–9.6 of riverine SPM), and a low POC:Chl *a* ratio (70–392 compared to 514–845 of riverine SPM except in March–April 2021. High Chl *a* concentrations, as well as a C:N ratio of 3–6 and POC:Chl *a* of <200, have been well established as biogeochemical proxies indicative of living phytoplankton [53,54]. In addition, much higher negative $\delta^{13}C$ values (−27.7 ± 0.2‰) and a much higher C:N ratio (40.6–125.5) of *P. australis* than those of R-SPOM indicate the minor contribution of the dominant marsh vegetation to the E-SPOM pool.

The compositional disparity in FAs between *P. australis*, R- and E-SPOM also suggests the minor role of riverine POM and marsh macrophytes in the E-SPOM pool. FAs are useful as source-specific biomarkers because it is often possible to identify unique FAs derived from different sources of organic matter (e.g., phytoplankton, detritus, vascular plants, terrestrial matter, and bacteria; Table S1). The high content of PUFAs (e.g., C18:3ω3, C20:4ω6, C20:5ω3, and C22:6ω3) in E-SPOM, compared to R-SPOM and *P. australis*, is indicative of the dominance of phytoplankton including diatoms [20,55], dinoflagellates [56], chlorophytes [57,58], and cryptophytes [37]. This result is consistent with the composition of dominant phytoplankton previously found in the Seomjin River estuary [29,31]. The contents of the long-chain SFAs C18:0 and C22:0—markers of nonliving detritus [59]—and C18:2ω6—a terrestrial-matter marker [55]—in E-SPOM were low, compared to those in R-SPOM. The much higher contents of C18:2ω6 (in all months) and C18:3ω6 (August 2020 and January 2021) in *P. australis* than those in R- and E-SPOM indicate a minor contribution of marsh macrophyte-derived organic matter to the POM pool. The higher ω3:ω6 ratio values—which represent a putative marker of the relative amount of terrestrial (<1) and aquatic (>1) sources [60]—in E-SPOM than those values in R-SPOM and *P. australis*, may also support the importance of phytoplankton to the E-SPOM pool. Overall, the evidence

obtained from biogeochemical proxies and FA biomarkers confirms that autochthonous production of phytoplankton explains most of the $\delta^{13}C$ and $\delta^{15}N$ values of E-SPOM.

*4.2. Importance of Estuarine Phytoplankton for the Corbicula Production*

Very narrow ranges of $\delta^{13}C$ and $\delta^{15}N$ values in clam tissues according to their size, site, and season were notable in the present study. This result may indicate consistency in *C. japonica*'s use of specified dietary resources. As indicated by the minor role of *P. australis*-derived organic matter in the E-SPOM pool, the negligible contribution *P. australis* to the nutrition of *C. japonica* can be determined by the very different $\delta^{13}C$ values. Given the great variation in the transport of riverine POM by freshwater discharge between the non monsoon and monsoon periods (Figure 1b), it may be expected that the availability of R-SPOM for the clam diets would vary with time. The isotopic similarity between R- and E-SPOM prevents separate assessment of the relative importance of each source to the *C. japonica* diets. However, as clearly indicated by biogeochemical proxies and FA biomarkers of SPOM, autochthonous phytoplankton-derived organic matter constitutes the main component of E-SPOM, and thus may account for the consistent isotope values of *C. japonica* in the present study.

The nMDS revealed that the FA profiles of the *C. japonica* tissues were closely associated with the E-SPOM. Indeed, the FA profiles of the *C. japonica* tissues were characterized by the dominance of a few PUFA compounds and the ω3:ω6 ratio, which are phytoplankton biomarkers (C18:3ω3 for chlorophytes, cryptophytes, and cyanobacteria; C20:5ω3 for diatoms; C22:6ω3 for dinoflagellates) (Table S1). In contrast, vascular plant (long-chain SFAs, C20 and C22) and bacteria (C18:1ω7) biomarkers were undetectable in clam tissues. Terrestrial-matter marker FAs (C18:2ω6 and C18:3ω3) were detected in the clam tissues, but these FAs are also dominant in chlorophytes. C18:2ω6 constituted a major class (7.0–28.7%) in *P. australis*, but only a minor content (0.4–2.3%) in the *Corbicula* tissues. The LINKTREE clearly showed that the FA profiles of the *C. japonica* tissues are linked to E-SPOM but separated from R-SPOM and *P. australis* by the criteria of the ω3:ω6 ratio and C20:5ω3. This result suggests the higher contribution of phytoplankton compared with riverine- and marsh-derived organic matter. This further indicates that phytoplankton are directly available from the water column for this filter-feeding clam. Despite the dominance of PUFAs in the *C. japonica* tissues, a slight seasonal variation in the FA profiles of the clam was attributed to the content of C22:6ω3, probably reflecting a seasonal succession of the phytoplankton community composition in our estuarine system [29]. The contents of C16:1ω7 (diatom biomarker) and C18:3ω6 (cyanophyte biomarker) were higher in E-SPOM than in R-SPOM or *P. australis*, and reversed with the season in the *C. japonica* tissues.

Dominant PUFAs (i.e., C18:3ω3, C20:5ω3, C22:6ω3, and C20:4ω6) were also found in other studies of *C. japonica* [61,62] and bivalves [63,64] (Table S2). These PUFAs are important for physiological functions in consumers and have been considered as essential dietary components [65]. C20:5ω3 and C20:4ω6 are believed to be related to osmoregulation and reproduction of bivalves [66]. A close association in the FA profiles between E-SPOM and *C. japonica* suggests that E-SPOM containing high proportions of PUFAs is more nutritious and plays a more important role in *C. japonica* nutrition than R-SPOM and *P. australis*. This explanation may be supported by the high %SPOM:SPM and molar C:N ratios in E-SPOM compared with R-SPOM or *P. australis*. In contrast, R-SPOM and *P. australis* were closely associated with C18:0, C22:0, and C18:2ω6, which are indicative of detritus, vascular plants, and terrestrial matter (Table S1). The proportions of these FAs were substantial in R-SPOM and *P. australis*, but found in only minor proportions in *C. japonica*, indicating that the former components are a minor contributor to the clam diet. Previous studies also found a high C18:2ω6 proportion in *P. australis*, but a paucity of this FA compound in bivalves [27,67]. The fact that C22:0 was undetectable in *C. japonica* tissues indicates that pedal (foot) feeding of vascular plant detritus did not occur.

Our understanding of the unique environmental features of the Seomjin River estuary and the distinct trophism of *C. japonica* may contribute to characterizing the low-turbidity

estuarine food web, and to restoring the ecosystem. The *C. japonica* population has been degraded since the late 1990s, when a gigantic steel mill and a waterway were constructed [68]. Based on our findings, it can be inferred that this decline in the *C. japonica* population could have been due to the interspecific competition with the invasive bivalve *Potamocorbula amurensis* for shared space and food after the anthropogenic disturbance [69]. One of the biological control methods may involve introducing *C. japonica* as natural competitors of the invasive species into the ecosystem to help control their populations. Another restoration effort may involve protecting overfishing and maintaining the *Corbicula* population density.

## 5. Conclusions

Previous SI studies have proposed the important roles of the riverine SPM in lowering estuarine ecosystems, especially for the *C. japonica* nutrition [9,42,43,70]. Unlike other *C. japonica* habitats, our SI and FA evidence suggest that autochthonous phytoplankton production of organic matter in the estuarine water column contributes most to *C. japonica* nutrition in the Seomjin River estuary. The dominant contribution of autochthonous phytoplankton production compared to riverine- and marsh-derived detritus to the E-SPOM pool was identified by their biogeochemical proxies and essential PUFA profiles. Close FA profiles between *C. japonica* and E-SPOM revealed that while the clam production is largely dependent on phytoplankton-derived organic matter (irrespective of clam growth, site, and season) in the unique low-turbidity estuarine condition, riverine- and marsh-derived particulate matter plays only a minor role in clam nutrition. Finally, the trophic role of terrestrial organic matter in the upper-estuarine ecosystem may vary, depending on regional conditions rather than generality, suggesting that the availability (quantity and composition) of R-SPOM and in situ primary production play a crucial role in determining the trophic base of an estuarine food web.

**Supplementary Materials:** The following supporting information can be downloaded at https://www.mdpi.com/article/10.3390/w15091670/s1, Table S1: Summary of FA biomarkers of putative carbon sources from literature; Table S2: Main/preferred carbon (food) sources inferred from fatty acid biomarkers of bivalve species; Table S3: One-way ANOVA was employed to test the significant difference between size groups; Table S4: FA composition (% of total FAs) of putative food sources [E-SPOM, R-SPOM, and *P. australis*] and results of two-way ANOVA based on main factors of the month and source type. The number of samples (*n*) is two for all cases. References [71–85] are cited in the Supplementary Materials.

**Author Contributions:** Conceptualization, D.S. and C.-K.K.; methodology, D.S. and C.-K.K.; validation, D.S., J.J. and C.K.; formal analysis, D.S. and D.K.; investigation, D.S., J.J. and C.K.; resources, D.S. and C.-K.K.; writing—original draft preparation, D.S.; writing—review and editing, C.-K.K.; visualization, D.S. and D.K.; supervision, D.S. and C.-K.K.; project administration, C.-K.K.; funding acquisition, C.-K.K. All authors have read and agreed to the published version of the manuscript.

**Funding:** This research was supported by the Korea Institute of Marine Science & Technology Promotion (KIMST) funded by the Ministry of Oceans and Fisheries (20220558).

**Institutional Review Board Statement:** Not applicable.

**Data Availability Statement:** Most data generated or analyzed during this study are included in this published article (and its supplementary information files). The dataset of environmental conditions analyzed during the current study is available from the corresponding author on reasonable request.

**Conflicts of Interest:** The authors declare no conflict of interest. The funders had no role in the design of the study; in the collection, analyses, or interpretation of data; in the writing of the manuscript; or in the decision to publish the results.

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
