# Peer review of "Identification of Phytoplankton-Based Production of the Clam Corbicula japonica in a Low-Turbidity Temperate Estuary Using Fatty Acid and Stable Isotope Analyses"

_water, doi:10.3390/w15091670_

Round 1

Reviewer 1 Report

General comments

Corbicula japonica is an important organism in the estuarine ecosystems, capable of removing large amounts of suspended matter from the estuary and it is a valuable bio-detector. It is also an important fishery resource due to its high secondary production. Therefore, much attention has been paid to its trophic ecology by the academic community.

While traditional analytical methods such as long-term direct observation in the field are time-consuming and labor-intensive to study, the biochemical tracer method is an efficient way to obtain information on the diet of target organisms over time by reflecting the assimilation of food sources in their tissues. Stable isotopes and fatty acids were chosen for the study as indicators for analysis. Carbon isotope ratio (d 13C) values of animal tissues reflect those of their diet and determine the carbon source of the animal. The nitrogen isotope ratio (d 15N) was used as a proxy for the nutritional status of the animal. FA enables the transfer of nutrients from producer to consumer to be tracked, distinguishing dietary fat from non-dietary fat. Thus, the above analysis leads to a reasonable determination of the trophic basis of the food web of the estuarine ecosystem.

The manuscript uses CTD to obtain water temperature and salinity data, and C. japonica was collected by sieving. Two collection sites for C. japonica and one downstream river site were selected as water sampling sites for isotopic and fatty acid (FA) measurements. It is scientifically valid to compare the differences between different sources at the same site. Elemental composition of C and N and FA biomarkers were analyzed using an elemental analyzer to distinguish between allochthonous and autochthonous food sources. The effects of clam size, salinity gradient and season were investigated to determine its resource use pattern in the low turbidity temperate Seomjin River estuary in Korea. The results suggest that C. japonica production under unique low turbidity estuarine conditions is heavily dependent on phytoplankton-derived organic matter, with river- and marsh-derived particulate matter playing only a minor role. Thus, the manuscript's study of the unique environmental and trophic characteristics of the Seomjin River estuary helped to characterize the food web of the low turbidity estuary in this region. Consequently, the manuscript can be accepted after major revisions.

Other problems are listed in detail as below

1.  The manuscript only describes the characteristics of the estuarine food web and lacks reflections and suggestions for ecosystem restoration. Please add relevant content.

2. Line 362: Here is only a brief description of the image. Please add a description of the image.

3. Line 159: It is the membrane that should be dried after the SPM measurement and not the filter used for the SPM measurement. Please clarify the formulation of the filter membrane.

4. Figure 5: The year and month are not clearly labelled in the chart, causing ambiguity in understanding. Please clearly indicate the year.

5.  Line11 and Line72: A technical abbreviation appearing at the first time should be marked with the full name.

6. Line24: Please standardize the keywords font.

7. Line 36 and Line 37: Delete the use of redundant dummy words, e.g. hence, therefore.

8. Table 1, Table 2, Table 4, Table S3, Table S4: Please mark the top corner with the information.

9. Table 3: Please add separators for numbers over 1000.

10. Line 192, Line 193, Line 200, Line 201: Please leave a space between the value and the unit.

Author Response

1. The manuscript only describes the characteristics of the estuarine food web and lacks reflections and suggestions for ecosystem restoration. Please add relevant content.

RESPONS: Thanks and agree. According to the reviewer’s suggestion, we have added some description relevant to ecosystem restoration. We have moved some descriptions of the ‘5. Conclusions’ section to last paragraph of the ‘4. Discussion’ section, as suggested by the reviewer #2, and combined discussion contents of interspecific competition with suggestions for ecosystem restoration. (see Lines 555-565 in the revised ms)

2. Line 362: Here is only a brief description of the image. Please add a description of the image.

RESPONSE: Thanks and agree. In this section, we explained a clear separation in FA compositions of E-SPOM, R-SPOM, and P. australis. Then we described the association in the FA profiles between food sources (i.e., E-SPOM, R-SPOM, and P. australis) and C. japonica on the nMDS plot (Figure 4) (see Lines 383-393). According to the reviewer’s comment, we have added more details explaining FAs contributing to this separation as follows: “This separation was attributed to close correlations of E-SPOM with w3:w6 ratio, C22:6w3, C20:4w6, and C20:5w3, and R-SPOM and P. australis with C22:0, C18:2w6, and C18:0”. (see Lines 364-366 in the revised ms)

3. Line 159: It is the membrane that should be dried after the SPM measurement and not the filter used for the SPM measurement. Please clarify the formulation of the filter membrane.

RESPONS: Thanks and agree. Accordingly, we have revised to clarify SPM measurement as follows: “Filters containing SPM were dried at 60°C for 72 h and the SPM mass was computed from the weight difference before and after filtration”. (see Lines 157-159 in the revised ms)

4. Figure 5: The year and month are not clearly labelled in the chart, causing ambiguity in understanding. Please clearly indicate the year.

RESPONSE: Thanks for checking and comment. As suggested by the reviewer, we have added “year” for all components in the revised figure. (see Figure 5 in the revised ms)

5. Line11 and Line72: A technical abbreviation appearing at the first time should be marked with the full name.

RESPONSE: Thanks. We made a mistake. Accordingly, we have revised “SIs” to “stable isotopes (SIs)”. (see Line 71 in the revised ms)

6. Line 24: Please standardize the keywords font.

RESPONSE: Thanks. We have fitted the font style and size in the revised version accordingly.

7. Line 36 and Line 37: Delete the use of redundant dummy words, e.g. hence, therefore.

RESPONSE: Agree. We have deleted “hence” and “therefore” in the revised ms accordingly

8. Table 1, Table 2, Table 4, Table S3, Table S4: Please mark the top corner with the information.

RESPONSE: Thanks for comment. This is likely to be an editorial issue.

9. Table 3: Please add separators for numbers over 1000.

RESPONSE: Agree. We have revised all in ‘Table 3’ accordingly.

10. Line 192, Line 193, Line 200, Line 201: Please leave a space between the value and the unit.

RESPONSE: Thanks for checking typo. We have revised according to the reviewer’s comment. (see Lines 199-200 in the revised ms)

Reviewer 2 Report

Reviewer’s Comments:

The manuscript “Phytoplankton-based production of the clam Corbicula japonica in a low-turbidity temperate estuary as identified by fatty acids and stable isotopes” is a very interesting work. This study presents a brackish water clam, Corbicula japonica, acts as an ecosystem engineer in estuaries. To identify its resource use patterns in the low-turbidity temperate Seomjin River estuary of Korea, we analyzed stable isotope and fatty acid (FA) biomarkers to differentiate allochthonous and autochthonous dietary sources and examined the effects of clam size, salinity gradient, and season. The 13C and 15N values were consistent across the three factors. The 13C values of clams were similar to those of both riverine- and estuarine-suspended particulate organic matter (R- and E-SPOM), while their 15N values were 2-4‰ higher, indicating an equal contribution of both sources to the clam diet. Biogeochemical proxies and FA compositions of SPOM indicates that estuarine phytoplankton significantly contributes to the E-SPOM pool. Moreover, the similarity in FA profiles between Corbicula and E-SPOM indicates that phytoplankton-derived organic matter is the primary source of nutrition for the clam, with minimal impact from growth, salinity gradient, or seasonal changes. The results are consistent with the data and figures presented in the manuscript. While I believe this topic is of great interest to our readers, I think it needs major revision before it is ready for publication. So, I recommend this manuscript for publication with major revisions.

1. In this manuscript, the authors did not explain the importance of the clam Corbicula japonica in the introduction part. The authors should explain the importance of clam Corbicula japonica.

2) Title: The title of the manuscript is not impressive. It should be modified or rewritten it.

3) Correct the following statement “Our findings indicated that allochthonous organic matter plays a minor role in clam nutrition in low-turbidity estuaries with high phytoplankton production compared to high-turbidity estuaries, which could help explain the differences in the trophic base of estuarine food webs across different regions”.

4) Keywords: The clam Corbicula japonica is missing in the keywords. So, modify the keywords.

5) Introduction part is not impressive. The references cited are very old. So, Improve it with some latest literature such as 10.3390/biom12010083, 10.1016/j.inoche.2022.109656

6) The authors should explain the following statement with recent references, “Although interaction effects between both factors were also observed in most of the FA classes (11 of 12), their importance varied among source types (Table S4)”.

7) Add space between magnitude and unit. For example, in synthesis “21.96g” should be 21.96 g. Make the corrections throughout the manuscript regarding values and units.

8) The author should provide reason for this statement “To understand those differences in SI values and to test our hypothesis, we applied FA composition to the identification of nutritional sources of C. japonica”.

9. Comparison of the present results with other similar findings in the literature should be discussed in more detail. This is necessary in order to place this work together with other work in the field and to give more credibility to the present results.

10) Conclusion part is very long. Make it brief and improve by adding the results of your studies.

11) There are many grammatic mistakes. Improve the English grammar of the manuscript.

Author Response

1) In this manuscript, the authors did not explain the importance of the clam Corbicula japonica in the introduction part. The authors should explain the importance of clam Corbicula japonica.

RESPONSE: Thanks for comment. We have described the importance of C. japonica as an ecosystem engineer, a biological filter affecting nutrient dynamics, and fisheries resources in the ‘Introduction’ section. (see Lines 28-38 in the revised ms)

2) Title: The title of the manuscript is not impressive. It should be modified or rewritten it.

RESPONSE: Thanks for comment. According to the reviewer’s suggestion, we have modified the title as follows: “Identification of Phytoplankton-Based Production of the Clam Corbicula japonica in a Low-Turbidity Temperate Estuary Using Fatty Acid and Stable Isotope Analyses”. (see Title in the revised ms)

3) Correct the following statement “Our findings indicated that allochthonous organic matter plays a minor role in clam nutrition in low-turbidity estuaries with high phytoplankton production compared to high-turbidity estuaries, which could help explain the differences in the trophic base of estuarine food webs across different regions”.

RESPONSE: Thanks for comment. According to the reviewer’s suggestion, we have modified this sentence in the ‘Introduction’ section as follows: We “Our study suggests that in low-turbidity estuaries with high phytoplankton production, allochthonous organic matter has a negligible contribution to clam nutrition compared to high-turbidity estuaries. This finding could provide insights into the variations in the trophic structure of estuarine food webs across diverse regions”. (see Lines 20-23 in the revied ms)

4) Keywords: The clam Corbicula japonica is missing in the keywords. So, modify the keywords.

RESPONSE: Thanks for comment. “Corbicula japonica” appear in the keyword list in the revised ms. (see Line 24 in the revised ms)

5) Introduction part is not impressive. The references cited are very old. So, Improve it with some latest literature such as 10.3390/biom12010083, 10.1016/j.inoche.2022.109656

RESPONSE: Thanks for critical comment. We have checked these papers. However, these papers have been published in ‘Biomolecules’ and ‘Inorganic Chemistry Communications’, respectively. These papers are not relevant to our study. We could not add thee references to our research list.

6) The authors should explain the following statement with recent references, “Although interaction effects between both factors were also observed in most of the FA classes (11 of 12), their importance varied among source types (Table S4)”.

RESPONSE: Thanks for comment. This is just our result, consistent with our studies. We already summarized FA biomarkers of different source types (i.e., source of organic matter or primary producers) in ‘Table S1’. Accordingly, as suggested by e reviewer, we have added ‘Table S1’ followed by ‘Table S4’ as follows: “(Table S4; see also Tables S1)”. (see Lines 350-351 in the revised ms)

7) Add space between magnitude and unit. For example, in synthesis “21.96g” should be 21.96 g. Make the corrections throughout the manuscript regarding values and units.

RESPONSE: Thanks for comment. Accordingly, we have carefully checked it throughout the manuscript and made corrections.

8) The author should provide reason for this statement “To understand those differences in SI values and to test our hypothesis, we applied FA composition to the identification of nutritional sources of C. japonica”.

RESPONSE: Thanks for this critical comment and agree. Accordingly, we have corrected in line with our intended meaning as follows: “To overcome this ambiguity in SI values and to test our hypothesis” (see Line 430 in the revised ms)

9) Comparison of the present results with other similar findings in the literature should be discussed in more detail. This is necessary in order to place this work together with other work in the field and to give more credibility to the present results.

RESPONSE: Thanks and agree. Based on stable isotope and fatty acid biomarkers, reports dealing with dietary sources of Corbicula japonica in estuarine systems are not common. We have referred to some references and make comparisons between different uses of terrestrial vs. estuarine sources of organic matter) in the ‘Discussion’ section. (see Lines 421-424; 439-441; 539-542 in the revised ms)

10) Conclusion part is very long. Make it brief and improve by adding the results of your studies.

RESPONSE: Thanks for comments. According to the reviewer’s suggestion, we have condensed the ‘Conclusions’ section. Instead, we have moved some descriptions of the ‘5. Conclusions’ section to last paragraph of the ‘4. Discussion’ section to add suggestions for ecosystem restoration (as also commented by the reviewer #1) and combine it with discussion content of interspecific competition (see ‘5. Conclusion’ in the revised ms)

11) There are many grammatic mistakes. Improve the English grammar of the manuscript.

RESPONSE: Thanks. We have improved our manuscript with the help of an available commercial company for English language editing services.

Round 2

Reviewer 1 Report

The authors have revised according to my suggestions and hopefully the manuscript is suggested to be accepted.